# Performance-Enhanced Static Modulated Fourier Transform Spectrometer with a Spectral Reconstruction

**DOI:** 10.3390/s23052603

**Published:** 2023-02-27

**Authors:** Ju Yong Cho, Seunghoon Lee, Won Kweon Jang

**Affiliations:** 1Department of Aeronautic Electricity, Hanseo University, 46, Hanseo 1-ro, Seosan-si 31962, Republic of Korea; 2Satellite Research Directorate, Korea Aerospace Research Institute, 169-82, Gwahak-ro, Daejeon 34133, Republic of Korea

**Keywords:** Fourier transform spectrometer, static modulation, spectral reconstruction, performance enhancement, transfer function

## Abstract

A static modulated Fourier transform spectrometer has been noted to be a compact and fast evaluation tool for spectroscopic inspection, and many novel structures have been reported to support its performance. However, it still suffers from poor spectral resolution due to the limited sampling data points, which marks its intrinsic drawback. In this paper, we outline the enhanced performance of a static modulated Fourier transform spectrometer with a spectral reconstruction method that can compensate for the insufficient data points. An enhanced spectrum can be reconstructed by applying a linear regression method to a measured interferogram. We obtain the transfer function of a spectrometer by analyzing what interferogram can be detected with different values of parameters such as focal length of the Fourier lens, mirror displacement, and wavenumber range, instead of direct measurement of the transfer function. Additionally, the optimal experimental conditions for the narrowest spectral width are investigated. Application of the spectral reconstruction method achieves an improved spectral resolution from 74 cm^−1^ when spectral reconstruction is not applied to 8.9 cm^−1^, and a narrowed spectral width from 414 cm^−1^ to 371 cm^−1^, which are close to the values of the spectral reference. In conclusion, the spectral reconstruction method in a compact static modulated Fourier transform spectrometer effectively enhances its performance without any additional optic in the structure.

## 1. Introduction

Concerning spectrometry, the Fourier transform spectrometer is interesting because it offers a high spectral resolution in the infrared region. Unlike diffractive optics-based spectrometry, it does not have the drawback of a diffraction limit in the infrared region; instead, the spectral resolution improves as the wavelength lengthens. A commercial Fourier transform spectrometer composed of a Michelson interferometer plays a critical role in many applications such as refineries, environmental monitoring [1], medical fields [2], space exploration [3], remote sensing [4], and other scientific areas [5,6,7,8,9,10,11,12]. To obtain a complete interferogram, the optical path difference between two divided beams at the beam splitter in the Michelson interferometer needs to be altered by moving one of the two mirrors back and forth. This is called dynamic modulation. The spectral resolution depends on the maximum optical path difference. Therefore, the mirror displacement needs to be long to achieve a long maximum optical path difference. Despite many advantages of Fourier transform spectrometers, the field application of dynamic modulated commercial Fourier transform spectrometers has been limited by strict operation prerequisites. They should be operated under a nitrogen environment to obtain clear spectral information and kept under high mechanical stability to protect the data acquisition from any vibration during operation.

The demand for fast measurement and immunity to vibration has led to the development of various static modulated Fourier transform spectrometers.

A static modulated Fourier transform spectrometer produces an optical path difference that is spatially distributed, resulting in an interferogram as a function of space. Some of these structures are reported as mirror-based interferometers [7,8,10,12,13,14] and birefringent prism-based interferometers [5,9,11,15,16]. However, the spectral resolution and spectral range of static modulated Fourier transform spectrometers can be worse than those of dynamic modulated Fourier transform spectrometers due to the limited number of data points, depending on the number of pixels of the detector. Moreover, the maximum optical path difference is also limited by the pixel pitch of the detector, resulting in a poor spectral resolution. Though the static modulated Fourier transform spectrometers have shown superior performance in terms of faster measurement of rapidly changing spectral properties of target substances, the main drawbacks of a worse spectral resolution and limited spectral range remain as major problems to be overcome to surpass dynamic modulated Fourier transform spectrometers.

A retarder array [11] and a stepped-mirror array [12,14] were introduced to extend the maximum optical path difference. A sensor shift method was suggested to enhance the spectral resolution [9]. However, these reports are based on utilizing additional optical components, which brings complications to a spectrometer. In addition, the spectral resolution was not comparable with those of dynamic modulated Fourier transform spectrometers. A signal-padding method was suggested to enhance the spectral resolution but it is only valid in limited conditions [10].

Recently, a static modulated Fourier transform spectrometer with multiple optical switches was reported [17,18]. An optical path difference was obtained by permutating pre-implemented optical switches. However, the data points were still insufficient and limited the spectral resolution and spectral range.

This drawback could be solved by employing a reconstruction method. In this study, a spectral reconstruction method was applied in a static modulated Fourier transform spectrometer composed of a modified Sagnac interferometer. After applying the reconstruction method to our static modulated Fourier transform spectrometer, the spectral resolution could be improved to a level comparable with that of a commercial Fourier transform spectrometer. 

Applying the reconstruction method requires the transfer function of a spectrometer, which is directly obtainable through an optical measurement of each wavenumber, performed by a monochromator. However, the spectral resolution depends on the performance of the monochromator. To overcome this dependency, we calculated the transfer function based on the relationship between an interferogram and a spectrum. The conditions for better spectral characteristics were investigated in terms of the ratio of a sampling frequency to the wavenumber of a source, mirror displacement, and the maximum optical path difference.

## 2. Methodology

A static modulated Fourier transform spectrometer was adopted that was composed of a modified Sagnac interferometer, as shown in Figure 1. SD is a source driver with a temperature controller, S is a radiation source, FL is a Fourier lens, and D is a one-dimensional array detector. A modified Sagnac interferometer consists of two mirrors, M1 and M2, and a beam splitter, BS. The two mirrors, M1 and M2, are tilted by 67.5° from the optical axis, and the beam splitter, BS, is tilted by 45° from the optical axis [8,13]. Radiation from the source splits into two beams at the BS, then they travel along each optical axis. If M2 was placed in a symmetrical position (the dashed line) where the displacement of M1 and M2 from the BS was equal, the two beams would have the same optical path length. In this case, an interferogram would not appear at the detector, D. However, if M2 was displaced by a distance, a, from the symmetrical position, the transmitted and reflected beams would be separated by *l*. An interferogram would be observed in an overlapped area of the two beams.

The separation distance between a transmitted and a reflected beam is given as follows:(1)$$l=\sqrt{2}a\;,$$
where l is the separation between a transmitted and a reflected beam, and a is the displacement of mirror M2 from the symmetrical point. Since a larger l provides a longer maximum optical path difference, l should be as long as possible for a better spectral resolution. Since the incident beam size to the detector is defined by the pupil size of the interferometer, and a larger l induces a narrower overlapped area width, the obtainable optical path difference must be limited. The maximum optical path difference and the effective sampling data points are limited by the separation of two beams, l.

In a static modulated Fourier transform spectrometer, a larger maximum optical path difference leads to an insufficient number of sampling data points, resulting in a poor spectral resolution and limited spectral range. Herein, we suggest using the spectral reconstruction method to solve this problem. It is assumed that an interferogram and a spectrum are linearly related as follows [18]:(2)I = TB,
where I is an interferogram, T is the transfer function, and B is spectral information. **I** and **B** are column vectors with *M* and *N* elements, respectively. **T** is a matrix with *M* × *N* elements. A detector is placed in a focal position to record a spatially distributed interferogram. An interferogram is expressed as the sum of cosine waves multiplied by spectral information associated with wavenumbers. Moreover, a cosine wave depends on parameters of a spectrometer such as the focal length of a lens, separation distance, and pixel pitch. We can assume that the transfer function of a spectrometer is interpreted as a cosine wave. Hence, Equation (2) is rewritten as follows:(3)$$\left(\begin{array}{c} I(1y_{\min})\\ \vdots \\ I(ny_{\min}) \end{array}\right)=\left(\begin{array}{ccc} \cos(2\pi {\bar{\nu }}_{\min}1y_{\min}\frac{l}{f}) & \ldots & \cos(2\pi {\bar{\nu }}_{\max}1y_{\min}\frac{l}{f})\\ \vdots & \ddots & \vdots \\ \cos(2\pi {\bar{\nu }}_{\min}ny_{\min}\frac{l}{f}) & \cdots & \cos(2\pi {\bar{\nu }}_{\max}ny_{\min}\frac{l}{f}) \end{array}\right)\left(\begin{array}{c} B({\bar{\nu }}_{\min})\\ \vdots \\ B({\bar{\nu }}_{\max}) \end{array}\right)\;,$$
where ${\bar{\nu }}_{\min}$ and ${\bar{\nu }}_{\max}$ are the minimum and the maximum wavenumbers, respectively. n is an integer representing the number of data points in the overlapped area, and $y_{\min}$ is the pixel pitch. Spectral reconstruction can be performed when an inverse matrix of the transfer function of a spectrometer exists.

Figure 2 shows the procedures used when applying the spectral reconstruction method. The transfer function was calculated with optical parameters such as lens focal length, pixel pitch, and mirror displacement. There was no phase information in the transfer function. A recorded interferogram contains phase errors due to thermal and electrical noise, sampling errors, and refraction, resulting in a shifted position of the zero path difference. Before the spectral reconstruction, a phase error should be corrected. We applied Mertz phase correction to the interferogram because it is commonly used in Fourier transform spectroscopy. Figure 2a shows the procedure for obtaining parameters for the transfer function of the spectrometer and phase information. The parameters and the phase information could be obtained from the spectrum. Since pixel pitch, focal length, and mirror displacement depend on the experimental setup, they were easily measured, but the focal length was slightly mispositioned due to refraction. To obtain an effective focal length, the spectrum must be obtained from the Fourier transform of an interferogram. The spectrum also contains a phase error that can be expressed as an angle along a wavenumber. The phase error can be calculated by the ratio of the imaginary part to the real part of the spectrum. Multiplying the calculated angle by the obtained spectrum compensates for the phase error. Since the center wavenumber of the radiation source was known, calibration could be performed and then the focal length calculated. Figure 2b shows the procedure for spectral reconstruction. An interferogram without any phase error could be obtained by inverse Fourier transforming a phase-corrected spectrum. The transfer function was calculated by Equation (3). Then the spectrum was reconstructed by calculating the residual sum of squares after rearranging Equation (2), given as follows [19,20]:(4)$$\mathrm{RSS}(B)={\left(T^{T}B-I\right)}^{2}\;,$$

A general standard regression was performed. However, the reconstructed spectrum still contained problems of overfitting and bias. For better reconstruction of the spectrum, an elastic-net method was employed [17,18]. This method takes advantage of both ridge and lasso regression so that overfitting and bias problems can be avoided.

A reconstruction method is valid when an interferogram is properly sampled according to the Nyquist sampling theorem and Shannon sampling criterion. Sampling frequency in a static modulated Fourier transform spectrometer is inversely proportional to the optical path difference corresponding to a single pixel that is given as follows:(5)$${\bar{\nu }}_{s}=\frac{1}{\Delta_{\min}}=\frac{f}{\sqrt{2}ay_{\min}}\;,$$
where ${\bar{\nu }}_{s}$ is the sampling frequency of a static modulated Fourier transform spectrometer, and $\Delta_{\min}$ is the optical path difference corresponding to a single pixel. To avoid aliasing, the sampling frequency must be at least twice the maximum wavenumber of the source.
(6)$$\varphi =\frac{{\bar{\nu }}_{s}}{{\bar{\nu }}_{\max}}\geq 2\;,$$
where φ is the ratio of the sampling frequency to the maximum wavenumber of the source, and ${\bar{\nu }}_{\max}$ is the maximum wavenumber of the source. The minimum wavenumber ${\bar{\nu }}_{\min}$ is 0. Associated with Equation (6), the spectral resolution depending on the width of the overlapped area created by mirror displacement is given as follows:(7)$${\bar{\nu }}_{R}=\frac{2({\bar{\nu }}_{\max}-{\bar{\nu }}_{\min})}{N_{\mathrm{overlap}}}=\frac{{\bar{\nu }}_{s}}{N_{\mathrm{overlap}}}=\frac{f}{\sqrt{2}aN_{\mathrm{overlap}}y_{\min}}=\frac{f}{(\frac{2r}{\sqrt{2}a}-1)2a^{2}}\;,$$
where ${\bar{\nu }}_{R}$ is the spectral resolution, $N_{\mathrm{overlap}}$ is the number of pixels in the overlapped area, and r is the incident beam radius to the detector. $N_{\mathrm{overlap}}$ is given by $\frac{A}{y_{\min}}$, and A is the width of an overlapped area given by 2*r*, which reduces with a.

## 3. Results and Discussion

The experiments were performed in a static modulated Fourier transform spectrometer composed of a modified Sagnac interferometer. In the experimental setup described in Figure 1, two square mirrors of 2 inches in width are silver-coated. A beam splitter with a transmittance of 50% in the wavenumber range from 3920 to 11,110 cm^−1^ is used. An LED is employed as the incident light source. The center wavenumber and the maximum optical power are 6451 cm^−1^ and 2 mW at a current of 0.55 A, respectively. A parabolic mirror is coupled with the LED to generate a parallel beam. In our study, a thermo-electric module was attached to the LED to control the temperature to 25 °C. At the width of the e^−2^ point from the highest intensity, the beam diameter was measured to be 8.23 mm. Pixel pitch and the number of pixels of the detector were 25 μm and 512, respectively. A 16-bit analog-to-digital converter was employed and the exposure time of the detector set to 4 ms. To compare the experimental data and the reconstructed results with the true spectral information, the reference spectrum was measured by a monochromator.

For successful spectral reconstruction, an interferogram should be recorded with the least distortion possible. Moreover, the ratio of the sampling frequency to the maximum wavenumber of the source needs to be considered. Figure 3 shows the optical path difference corresponding to a single pixel and the ratio of the sampling frequency to the maximum wavenumber φ along mirror displacement a. The focal length of the Fourier lens was 300 mm. As mirror displacement a increases, the optical path difference $\Delta_{\min}$ linearly. The ratio φ decreases along a and falls below two when a is greater than 6.3 mm. When the ratio φ is below two, an interferogram is insufficiently sampled, leading to aliasing in the spectrum. a may be no longer than 6.3 mm to avoid aliasing.

As a increases, the distance l increases proportionally. As a result, the length of an overlapped area decreases, leading to a decrease in sampling data points. Figure 4 shows the variation of data points in an overlapped area and the spectral resolution along mirror displacement. As the mirror displacement increases, the optical path difference corresponding to single pixel $\Delta_{\min}$ increases so that the maximum optical path difference increases as the spectral resolution is enhanced. However, as the length of an overlapped area decreases, the maximum optical path difference begins to decrease, resulting in deterioration of spectral resolution. In this experiment, the longest maximum optical path difference was obtained at a mirror displacement of 5 mm. The best spectral resolution was expected to be 74 cm^−1^. Moreover, since φ was close to two, it was highly possible to obtain an interferogram with the lack of sampling data points.

As shown in Figure 4, the static modulated Fourier transform spectrometer with a modified Sagnac interferometer has a poor spectral resolution, similar to other reports. However, through a reconstruction method, the spectral resolution of the system can be significantly improved. When the reconstruction method is applied to a measured spectrum, the spectral resolution becomes dependent on the number of wavenumber components in the transfer function. In this experiment, the wavenumber range of the transfer function is from 3584 to 12,472 cm^−1^, which is coincident with that of the detector, and the number of components is 1024, resulting in a spectral resolution of 8.7 cm^−1^ at any mirror displacement. The number of components should be 1024 at most because the spectrum cannot successfully be reconstructed when the number of components is greater than 1024. This is because the complexity inducing the bias and overfitting problem grows as the number of components increases. In addition, to obtain the narrow spectral width, an interferogram needs to be sampled for a long maximum optical path difference according to the Shannon criterion. The optimal mirror displacement should be decided by careful consideration. Figure 5 shows interferograms and Fourier transformed spectra along mirror displacement. Figure 5a is an interferogram obtained at a mirror displacement of 1 mm. The number of data points in the overlapped area is 457 and the maximum optical path difference is 5.6 × 10^−2^ mm. The maximum optical path difference $\Delta_{\min}$ is determine by multiplying $N_{\mathrm{overlap}}$. Figure 5b shows the spectrum at a mirror displacement of 1 mm. The monochromator (black solid line) is the reference spectrum. F.T. (black dashed line) is a Fourier transformed spectrum. When Fourier transform is performed, the spectral width is 596 cm^−1^ and the spectral resolution is 178 cm^−1^. However, after the reconstruction method is applied, the spectral width reduces to 468 cm^−1^ and the spectral resolution improves to 8.9 cm^−1^. The spectral resolution could be significantly improved but the spectral width was still wider compared to that of the reference spectrum due to the small value of the maximum optical path difference. We could obtain spectra along mirror displacement by performing Fourier transform and reconstruction to find the condition under which the spectral width became the narrowest. The best spectral resolution of 73 cm^−1^ was obtained at a mirror displacement of 5 mm and a spectral width of 428 cm^−1^. However, in this experiment, the narrowest spectral width was obtained at mirror displacement of 4 mm. The difference might have arisen from sampling errors that broadened the spectral width, with the effects of the errors dominant when φ became closer to the condition in Equation (7). At the mirror displacement of 4 mm, φ was 3.2, which was greater than the 2.6 obtained for a mirror displacement of 5 mm. Therefore, the optimal mirror displacement should be adjusted by considering the optimal value of φ. Figure 5c is an interferogram obtained at a mirror displacement of 4 mm. The number of data points in the overlapped area is 288, and the maximum optical path difference is 13.6 × 10^−2^ mm. Figure 5d shows the spectrum at a mirror displacement of 4 mm. When Fourier transform is performed, the spectral width and spectral resolution are 414 cm^−1^ and 74 cm^−1^, respectively. When the reconstruction is performed, the spectral width reduces to 371 cm^−1^ and the spectral resolution improves to 8.9 cm^−1^. The reconstructed spectrum shows much closer characteristics to the reference spectrum.

Figure 5e is an interferogram obtained at a mirror displacement of 6 mm. Though mirror displacement increases, the spectral width becomes broader than that at a mirror displacement of 4 mm. The number of data points in the overlapped area is 175, and a maximum optical path difference is 12.3 × 10^−2^ mm. Compared to Figure 5c, the number of data points decreases and the maximum optical path difference shortens. Figure 5f shows a spectrum at a mirror displacement of 6 mm. When Fourier transform is performed, the spectral width is 450 cm^−1^ and the spectral resolution is 81 cm^−1^. When the reconstruction is performed, the spectral width reduces to 407 cm^−1^ and the spectral resolution improves to 8.9 cm^−1^.

Figure 5g shows a spectral width and φ along mirror displacement. A black solid rectangle and a hollow rectangle show the spectral width obtained by performing Fourier transform and reconstruction, respectively, along mirror displacement. The hollow circle is φ. The spectral width of the spectrum obtained by the reconstruction method is narrower than that obtained by Fourier transform. Moreover, the reconstructed spectrum is more accurate when the maximum optical path difference is long in an obtained interferogram. The maximum optical path difference is found for the longest mirror displacement of 5 mm; however, the spectral width is not the narrowest. It should be considered that, for the best performance of the spectrometer, not only the maximum optical path difference but also the sampling frequency should be considered. After correction of sampling errors, the optimal φ value of 3.2 was obtained at a mirror displacement of 4 mm.

## 4. Conclusions

A static modulated Fourier transform spectrometer is composed of a modified Sagnac interferometer. Due to insufficient sampling data points in the overlapped area, the maximum wavenumber is limited, resulting in distortion such as spectral folding. A modified Sagnac interferometer is readily optimized to overcome the distortion, thus improving on other static modulated interferometers such as a single mirror-based interferometer and a birefringent prism-based interferometer.

In this investigation, from the relationship between the interferogram and the spectrum, the transfer function of a spectrometer could be expressed as a cosine function. Important parameters of the transfer function were focal length of the lens, mirror displacement, and pixel pitch. In spectral reconstruction, the spectral resolution depends on the number of wavenumber components of the transfer function. The spectral resolution could be improved to 8.9 cm^−1^ at any mirror displacement when the number of wavenumber components in a transfer function was 1024 in the range from 3584 to 12,472 cm^−1^. This marked a significant improvement compared to the value of 74.0 cm^−1^ obtained by performing Fourier transform. To obtain better spectral characteristics, a larger maximum optical path difference is required. Thus, the ratio of the sampling frequency of a spectrometer to the maximum wavenumber should be considered. In our experiment, when φ became closer to two, the errors dominantly affected the spectrum, and as a result, the spectral width was broadened. The best condition for reconstruction was at a mirror displacement of 4 mm, where φ became 3.2. Before applying the reconstruction method suggested in this study, the obtained Fourier transformed spectral width was 414 cm^−1^; then, after the reconstruction method was applied, the spectral width was narrowed to 371 cm^−1^, which was very close to the reference spectral value. The spectral reconstruction was successful in improving the spectral characteristics in a static modulated Fourier transform spectrometer. The spectral resolution could be greatly improved without requiring any additional optical component to increase the maximum optical path difference. However, the accuracy of the spectrum is still dependent on the maximum optical path difference and sampling frequency. An approach to solving this drawback needs to be discussed. The reconstruction method that we suggest in this paper allows a static modulated Fourier transform spectrometer to remain compact with high performance.

## Figures and Tables

**Figure 1 sensors-23-02603-f001:**
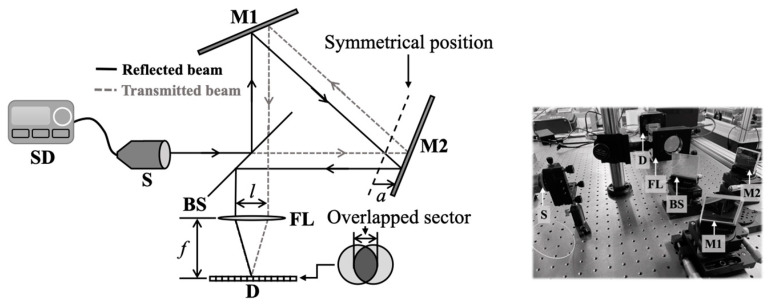
A Fourier transform spectrometer composed of a modified Sagnac interferometer.

**Figure 2 sensors-23-02603-f002:**
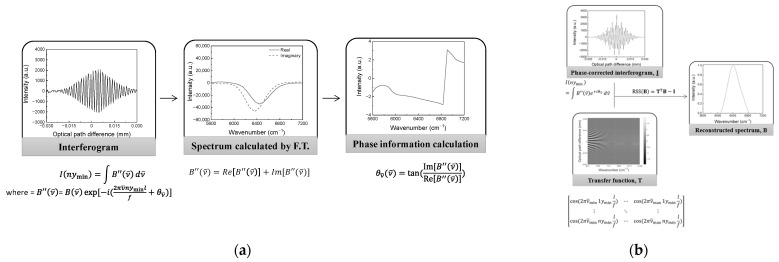
Spectral reconstruction procedure. (**a**) Procedure used to obtain phase information and parameters of the transfer function. (**b**) Reconstruction procedure.

**Figure 3 sensors-23-02603-f003:**
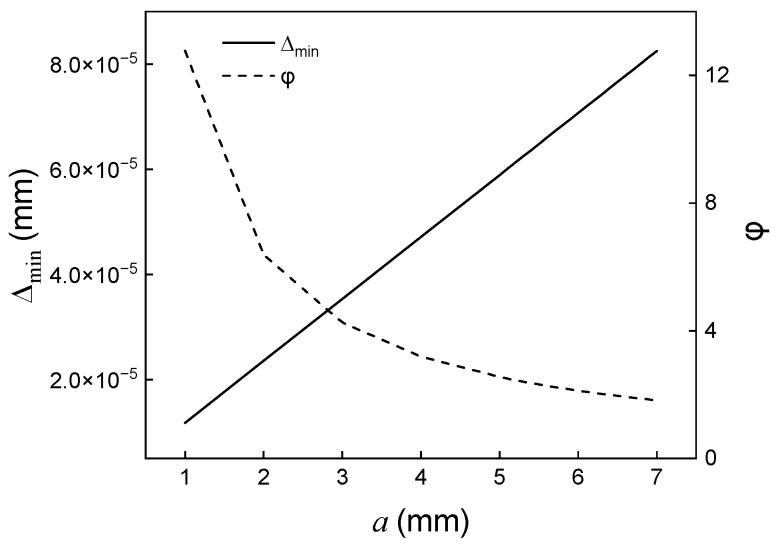
Variation of optical path difference of a single pixel and the ratio of sampling frequency to maximum wavenumber of source radiation along mirror displacement.

**Figure 4 sensors-23-02603-f004:**
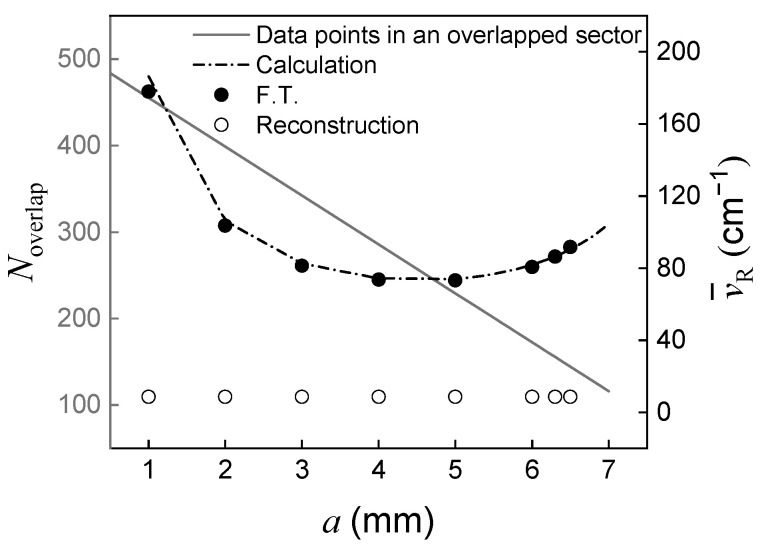
Variation of sampling data points in an overlapped area and spectral resolution along mirror displacement.

**Figure 5 sensors-23-02603-f005:**
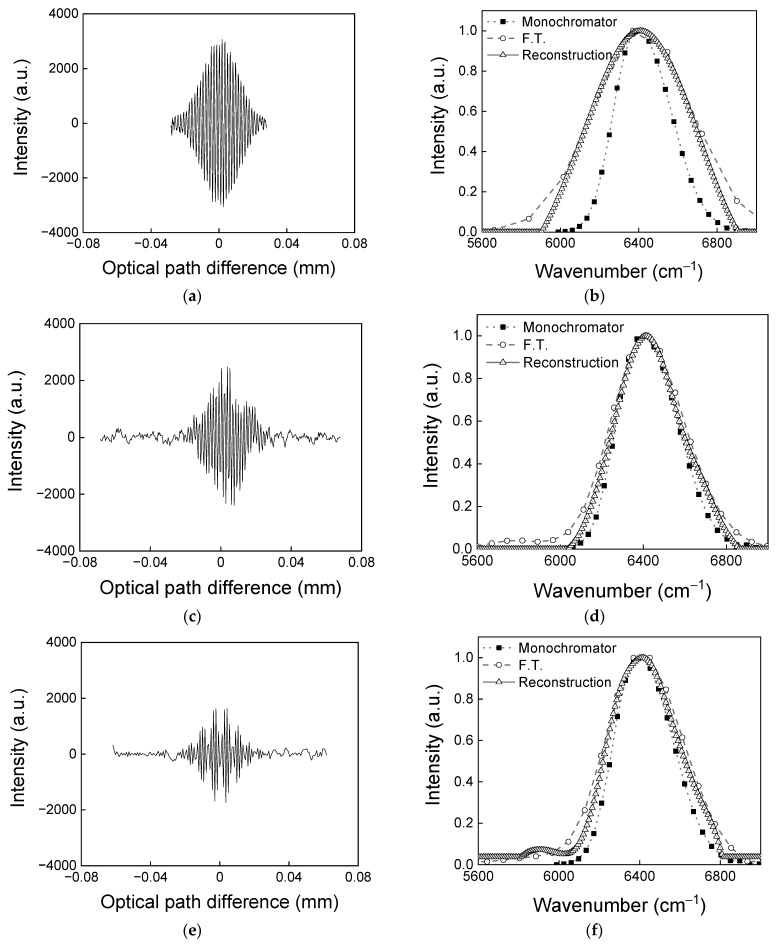
An interferogram and spectra along with mirror displacement. (**a**) An interferogram obtained at a mirror displacement of 1 mm, (**b**) spectra corresponding to (**a**), (**c**) an interferogram obtained at a mirror displacement of 4 mm, (**d**) spectra corresponding to (**c**), (**e**) an interferogram obtained at a mirror displacement of 6 mm, (**f**) spectra corresponding to (**e**), and (**g**) the spectral width associated with the method and φ, along with mirror displacement..

## Data Availability

Not applicable.

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
