# Peer review of "Performance-Enhanced Static Modulated Fourier Transform Spectrometer with a Spectral Reconstruction"

_sensors, 2023, doi:10.3390/s23052603_

Round 1
Reviewer 1 Report
This paper “Performance-enhanced static modulated Fourier transform spectrometer with a spectral reconstruction” aims to introduce an enhanced performance of a static modulated Fourier transform spectrometer with a spectral reconstruction method that can compensate for the insufficient data points. The authors analyzed the relationship between a measured interferogram and a transformed spectrum, then calculated a transfer function from measured spectral information. The spectral reconstruction has been performed by linear regression for the transfer function and the interferogram. In a modified Sagnac interferometer structure, the ratio between a sampling frequency of spectrometer points and the maximum wavenumber, the spectral resolution, and the spectral width could be calculated and analyzed along the mirror displacement that generated the optical path difference. At an optimal experimental condition, by performing spectral reconstruction, a spectral resolution could be drastically improved from 74 to 8.9 cm−1, and a spectral width narrowed from 414 cm−1 to 371 cm−1, which is close to that of the reference spectral value. Authors also claimed that the spectral reconstruction method, in a compact static, modulated Fourier transform spectrometer, is effective to obtain a highly enhanced performance.
The topic is justified. The paper could be further improved if the following remarks are taken into consideration:
1. ABSTRACT: The text should include more details about the proposed methodology, numerical results achieved, and a comparison with other methods.
2. Few grammatical mistakes were found in the whole draft of the article; the authors need to fix these.
3. Introduction section seems ok to hold the rationale and proper justification of the study.
4. The study should also discuss some recently published related (conducted in the year 2021-2022), to conclude a proper research gap and establish a research problem.
5. Better to write an algorithm of the proposed methodology also with its time complexity analysis.
6. Although, the results are convincing, however, they should be compared to at least one state-of-the-art study.
7. Confused, why did the authors set the exposure time of the detector to 4m?
8. The motivation is not clear. Please specify the importance of the proposed solution.
9. Discuss the limitations of the proposed method with their possible solutions in the future work section.
10. Conclusion section may be comprehensive.
11. Most of the references are not relevant to the conducted experiments.
Author Response
The authors thank the reviewers for their careful reading of the manuscript and constructive remarks. And we have taken into account the comments to improve and clarify the manuscript. Please find below a detailed point-by-point response to all comments. The reviewer’s comment is in blue, and our respond is in black.

Reviewer 2 Report
The manuscript "Performance-enhanced static modulated Fourier transform spectrometer with a spectral reconstruction" by Cho et al. introduced a method of spectrum reconstruction for the static Fourier transform spectrometer.
The authors claimed that the spectral resolution can be significantly improved by the reconstruction method. I think the evidence for such claim is insufficient. To prove such spectral resolution (say 8 cm-1), the author should show the proof that two features that are 8 cm-1 apart can be successfully resolved with the reconstruction method. The experiments in the manuscript only shown the different shape before or after the reconstruction. While the latter did become more similar to the reference, it is not a solid proof of the resolving power for the method.
Also, since the method involves a focusing optics, I would like to see how different abberations (chromatic, astigmatism etc.) would be dealt with in the method.
Overall, I would not suggest to publish this manuscript until proper experimental evidence for the resolving power are shown.
Author Response

(The authors gave the same response as above.)

Reviewer 3 Report
The article "Performance enhanced static modified Fourier transform spectrometer with a spectrum reconstruction" described a static modified Fourier transform spectrometer. However, there was a lack of detailed discussion on the strategy, implementation process and mechanism of the performance enhanced method, which need to be supplemented.
1. FL is not a focusing lens, but a Fourier lens. The radiation source S should be located at its object focus, and the detector should be located at its image focus.
2. Because I=TB, according to the restriction of matrix dimension, the transpose of T and B cannot be multiplied, unless T is a square matrix, so TTB-I should not be work in RSS (B)=(TTB-I)2 of formula (4).
3. How to get the last step of formula (7)? Please supplement the detailed derivation process.
4. A parabolic mirror is coupled with the LED to generate a parallel beam, is this parabolic mirror the focus lens FL of Figure 1?
5. What material is used for the detector and what is its spectral response range? InGaAs or not?
6. In Figure 3, please give the analytic expression of Δmin=f(a) and φ=f(a).
7. In Figure 4, please give the analytic expressions of Noverlap=f (a) and vR=f (a).
8. In line 218, the number of detector components is 1024, but in line 182, the number of detectors is 512, which is inconsistent.
9. Give physical photos of the experimental device and the experimental site.
10. Curve of spectral width and spectral resolution along with mirror displacement during experiment should be supplemented.
11. The description of the Reconstruction method is too little. The mechanism of Performance enhanced method and the specific implementation strategy are not described. The detailed process need to be supplemented.
Author Response

(The authors gave the same response as above.)

Round 2
Reviewer 2 Report
In the authors response, unfortunately, neither of my concerns are sufficiently addressed.
1. The authors claim that, to reveal two peaks 8 cm-1 apart, a spectral resolution finer than 8 cm-1 is required. If so, I would like the authors to comment on what would be the closest peak distance that an 8 cm-1 resolution can resolve and provide some experimental evidence on that. I understand that finer spectral resolution can give narrower FWHM. But this, at least to me, was insufficient to demonstrate the capability of information extraction for this reconstruction method. As the Rayleigh criterion for imaging is defined by resolving two adjacent points, I would like to see direct evidence of resolving two adjacent peaks, which are not resolvable prior to applying the reconstruction method.
2. Mertz correction is designed to correct phase error in FTIR. While converting the dynamic FTIR to static, which involves an imaging process, phase error is not the only error that exists. Due to aberration, the signal amplitude can be different from a perfect imaging system (blurring, for example, the amplitude at one point would be influenced by adjacent points). This is what I would like the authors to address and see if the reconstruction method can do any improvement on that.
Author Response
The authors thank the reviewers for their careful reading of the manuscript and constructive remarks. We have carefully taken into account the comments to improve and clarify the manuscript. Please find below a detailed point-by-point response to all comments. The reviewer’s comment is in blue, and our response is in black.

Reviewer 3 Report
Although the author revised the manuscript according to the review comments, the revision was not detailed, and the revised manuscript did not make detailed additions according to the revision comments, such as the photos of the experiment site, etc. In addition, the revised draft is uploaded in the revision mode, and the format is messy. At the same time, the coordinates in some figures are "?", indicating that the author does not pay enough attention to the revision comments. The author is expected to revise carefully to meet the modification requirements.
Author Response

(The authors gave the same response as above.)

Round 3
Reviewer 2 Report
The author provided two references for the spectral resolution, which I greatly appreciate. In both references provided by author, the spectral resolutions are determined by the FWHM of one or several spectral features.
However, in these references, the FWHM is similar in number to the spectral resolution authors claimed. While in this manuscript, the authors claim <10 cm-1 spectral resolution when showing a peak with FWHM ~400 cm-1. I understand the feature itself can be wide, but this appears to be not convincing enough.
Thus I suggest either showing a feature with FWHM similar to the claimed spectral resolution after reconstruction, or showing the spectral resolving power by resolving two adjacent features that's 9 cm-1 (or the smallest number achievable by claimed resolution) apart. Otherwise the authors are claiming too aggressively.
To further clarify, I never required to use light sources that have two adjacent peaks in spectrum. I understand that for an FTIR, the light source should be wide spectrum. For all the features I mentioned above, I meant the feature of a sample/specimen.
Author Response
We have carefully taken into account the comments to improve and clarify the manuscript. Please find the attached file of our response.

Reviewer 3 Report
Accept
Author Response
We are very pleased to have satisfied the reviewer with our responses.